# A High-Precision Voltage-Quantization-Based Current-Mode Computing-in-Memory SRAM

**DOI:** 10.3390/mi14122180

**Published:** 2023-11-29

**Authors:** Ruiyong Zhao, Zhenghui Gong, Yulan Liu, Jing Chen

**Affiliations:** 1Shanghai Institute of Microsystem and Information Technology, Chinese Academy of Sciences, Shanghai 200031, China; zry@mail.sim.ac.cn (R.Z.); ylliu@mail.sim.ac.cn (Y.L.); 2University of Chinese Academy of Sciences, Beijing 100049, China

**Keywords:** SRAM, computing-in-memory, analog non-ideality, high-precision fully dynamic range IV conversion circuit

## Abstract

Non-linear distortion of signals is a serious problem in computing-in-memory SRAM (CIM-SRAM) circuits in current mode. This problem greatly limits the performance of calculations and directly affects the computing power of the CIM-SRAM. In this study, the causes of non-linearity and inconsistency were investigated. Based on detailed analyses, we proposed a high-precision, fully dynamic range IV (HFIV) conversion circuit. The HFIV circuit was added to each bit line (BL) for voltage clamping and proportionally mirroring the read current. We applied the structure to numerous prior studies and evaluated them using the 55 nm complementary metal-oxide semiconductor process. The results showed the proposed HFIV circuit could increase the CIM-SRAM’s calculation linearity to 99.92% (8~32 SRAM bit-cells) and 99.8% (32~64 SRAM bit-cells) with a 1.2 V supply.

## 1. Introduction

In today’s post-Moore era, the data transfer process is deeply limited by the storage wall, leading to the development speed of memory seriously lagging behind the processor, and thus making the computing power of chips difficult to improve. The integration of memory and computing breaks the mainstream traditional paradigm of a von Neumann computing architecture. Its computing architecture turns to data-centric, embedding computing power in memory, and supports parallel computing. It can minimize the energy consumption and time delay caused by data handling, and supports large-scale parallel operations. Therefore, the power consumption, bandwidth, and speed of the in-memory computing architecture have been improved to varying degrees. SRAM based on in-memory computing macros is a high-speed, energy-efficient hardware ideal for edge devices that deploy neural network algorithms. Deep neural network models have made many pioneering achievements in various edge computing applications, such as face recognition, image processing, speech recognition, etc. Binary neural networks, with their potential advantages of a high model compression rate and computing speed, have become a popular research direction for mobile AI Internet of Things. Among them, the MAC operation occupies almost 95% of the neural network model convolution algorithms and consumes most of the computational resources. Therefore, how to achieve a high accuracy of the MAC operation is the key to memory computing [1]. 

CIM-SRAM’s calculation modes can be divided into four types: current mode, voltage mode, charge-sharing mode, and capacitive-coupling mode [2]. Current, charge-sharing, and capacitive-coupling modes can achieve high parallelism and high precision. The cell structure in current mode is the simplest of all the CIM arrays. Current mode only requires 2–4 transistors for each bit-cell to finish multiplication or a Boolean operation. By using Kirchhoff’s current law, the multi-bit multiplication result accumulating operation can be completed. Compared with the voltage-mode memory calculation, no additional adder circuit is required, which needs a large area. At the same time, compared with charge-sharing mode and capacitive-coupling mode, no capacitance structure needs to be added to each unit, so it has a smaller unit and array area and higher area efficiency ratio. However, there are still two key problems in the SRAM macro based on the existing current-mode memory computing mode [3]: (a) when there are large numbers of units, the current accumulation may cause simulation non-linearity and result in fewer bits and (b) the limited dynamic range may cause the signal margin to decrease [3].

These two problems greatly limit the performance of the calculation and directly affect the computing power of the whole circuit. In the MAC operation, increasing the number of inputs and weights or accumulation results in an exponential decrease in the signal margin and the outcome linearity. However, when the signal margin is less than the input offset of the analog readout circuit, the sensing amplifier (SA) or the ADC, the accuracy of the analog readout operation, decreases, resulting in reduced system-level calculation accuracy. Recently, several studies [4,5,6] identified this problem in the SRAM design and, therefore, designed multiple methods to alleviate non-linear current accumulation. For example, in [5], a current mirror is added to improve the linearity by accelerating the BL discharge in the triode region. However, excessive discharge operations may occur with overcompensation or read breaking. A double 8T (T8T) SRAM unit for multiple MAC calculations was proposed by Si et al. [6]. Their CIM macros support configurable operations of up to 4b inputs, 5b weights (1b symbols, 4b data), and 7b outputs. Two standard 8t structures were used, which support multi-bit convolutional neural networks. But, when the number of units along the BL increases, the increased leakage current may reduce the linearity and the accuracy of the calculation. The current general measure [7] is to set the dynamic range of the signal within a limited interval, to improve the linearity of the current accumulation results and ensure the accuracy of the operation results. This practice cannot achieve the signal margin of rail to rail, thus limiting the maximum calculation accuracy. 

In the design process, the linearity and accuracy need to be balanced according to the index. Compared with the complex high-precision ADC structure (most high-precision ADC structures cannot avoid the complex circuit structure and large power consumption problems), the high-precision voltage gradient quantification structure proposed in this paper is different from the quantification of most ADC structures on the market and can effectively reduce the area and power consumption cost.

This work proposes a high-precision voltage gradient quantization SRAM macro based on 6T SRAM, as shown in Figure 1. The multiplication data come from the binary image data input outside of the array and the binary weight data stored in the array. The input image and the weight data are subjected to the multiplication and addition MAC operation in the analog domain in the integrated storage array, which has higher computational efficiency and lower power consumption without losing the accuracy of the binary operation. In the accumulation calculation, the current accumulation method is used to add the multiplication current results obtained by each storage unit into the analog current output. Through the HFIV circuit, as shown in Figure 2, the output current of each cycle will be converted into a quantized steady-state voltage output. Finally, the gradient voltage decoder converts the data from the analog domain to the digital domain. Compared with the complex high-precision ADC structure, HFIV-SRAM’s area and power consumption overhead are reduced. The voltage gradient quantization SRAM macro is composed of a 64 × 64 memory computing array, eight column-level HFIV modules, and a gradient voltage decoder. The process used is SMIC 55 nm. The simulation results show the output accuracy of the voltage gradient quantization SRAM macro is 5 bit and the operating frequency is 200 MHz. The contribution of this work is as follows: a HFIV circuit is designed to overcome the problem that the output accuracy is greatly reduced due to the non-linear phenomenon caused by multi-line reading technology.

## 2. Challenges of Current-Based CIM-SRAM

### 2.1. Analysis of Analog Non-Idealities

The 8T SRAM structure proposed in [8] can be used as a typical current-mode memory computing unit to simulate non-ideal row analysis. The structure is shown in Figure 3. In the operation of the structure, the accumulated results are multiplied by the units in the same column to produce or not produce current. The current accumulates in the read-bit-line (RBL) to produce the RBL/RBLB’s voltage drop (∆VBL/∆VBLB) representation, as shown in Figure 4. To obtain a higher linearity for the current accumulation results, a limited dynamic voltage range is usually set.

When the structure accumulates current, the analog–digital conversion is usually performed at the end of the RBL. Taking the diode load in Figure 5 as an example, the multiplicative calculation current generated by each IMC unit can be expressed as: (1)$$I_{ac}\approx \frac{1}{2}\mu_{n}C_{ox}\frac{W_{c}}{L_{c}}{\left(V_{DD}-V_{\mathrm{RBLB}}-V_{\mathrm{TH}}\right)}^{2}(1+\lambda V_{\mathrm{DS}})=A_{C}{\left(V_{\mathrm{DD}}-V_{\mathrm{RBLB}}-V_{\mathrm{TH}}\right)}^{2}(1+\lambda V_{\mathrm{DS}})$$

For the load diode NMOS, the current *I_T_* flowing through the NMOS is represented as follows: (2)$$I_{T}=I_{\mathrm{ac}}\approx \frac{1}{2}\mu_{n}C_{ox}\frac{W_{T}}{L_{T}}{\left(V_{\mathrm{RBLB}}-V_{\mathrm{TH}}\right)}^{2}(1+\lambda V_{\mathrm{DS}})=A_{T}{\left(V_{\mathrm{RBLB}}-V_{\mathrm{TH}}\right)}^{2}(1+\lambda V_{\mathrm{DS}})$$

The mathematical relationship between *I_c_* and *I_T_* is: (3)$$I_{T}=\sum_{i=1}^{N}I_{c}=N\cdot I_{c}$$

The function relationship between the total current of the RBL and the number of units can be obtained by solving: (4)$$I_{T}=A_{T}\frac{{\left(V_{\mathrm{DD}}-2V_{\mathrm{TH}}\right)}^{2}}{{\left({\left(\frac{A_{T}}{{\mathrm{NA}}_{C}}\right)}^{\frac{1}{2}}+1\right)}^{2}}$$

Equation (4) is not a linear relationship. It can be seen from Figure 6 that, due to the non-linearity transconductance characteristics of the load, the current IT on the RBL accumulates when the number of units n is larger, and the non-linearity becomes more serious. Therefore, in the design of the current mode, the number of units accumulating current at the same time should be minimized to achieve relatively better linearity, but this will lead to a decrease in the parallelism of the operation and a decrease in the number of calculation bits.

### 2.2. Analysis of Dynamic Range and Signal Margin

#### 2.2.1. Standard 6T Unit without Read Isolation Structure

The standard 6T SRAM cell without a read isolation structure [9] is shown in Figure 6. The standard 6T SRMA cell obtains different accumulated currents on the BL by inputting different word line (WL) voltages to generate a BL voltage drop for the multiplication operation in the analog domain, and simultaneously opens multiple sets of word lines on the BL to complete the accumulation operation. To improve the energy efficiency, a large number of word lines will be activated at the same time to increase the accumulation. When the number of activated WLs causes the BL voltage swing range to exceed the unit’s flip threshold voltage, it will lead to the phenomenon of read interference, as shown in Figure 7. To avoid the problem of read interference, the voltage swing range of the bit line (*V_BLSR_*) needs to be maintained between the unit data flip threshold voltage and the power supply voltage. The *V_BLSR_* limits the maximum number of WLs activated at the same time and the voltage signal margin calculated by analog accumulation, resulting in a decrease in the number of calculated bits.

#### 2.2.2. Computing-in-Memory SRAM Cell with Read Isolation Structure

The 8T SRAM structure proposed in [10] has a read–write isolation structure, which can avoid the unit storage data flipping when the unit is calculated. In the design of this structure, the signal margin depends on the conversion accuracy of the digital-to-analog converter. When a large number of units are activated in the same column, the non-linear increment of *I_AC_* leads to a smaller conversion voltage increment, and the requirement for ADC will be greatly increased. When too many word lines are activated at the same time, the resulting RBL voltage drop ∆VBBL/∆VRBLB will be lower than the minimum voltage gradient that the ADC can accurately convert. At this time, a digital-to-analog conversion error will occur. Figure 8 illustrates the error rate of analog-to-digital conversion increases with the increase in the number of activated word lines.

## 3. Design and Analysis

### 3.1. High-Precision Full Dynamic Range IV (HFIV) Conversion Circuit 

To solve these two problems of fewer bits caused by analog non-ideality and the decrease in the signal margin caused by a limited dynamic range, a high-precision fully dynamic range IV (HFIV) conversion circuit is proposed, which provides a higher precision and larger dynamic range analog quantization voltage to the later coding circuit. The HFIV circuit has an improved traditional current mirror structure, which consists of two operational amplifiers, an LVT bias transistor, and a voltage quantization resistor, as shown in Figure 9. 

For the current isolated calculation structure of the traditional current mirror load, there are two problems that lead to current accumulation mismatch and replication mismatch (Figure 9): (1) the voltage VRBL on the current accumulation bit line and the drain voltage mismatch of the current replication transistor lead to replication current mismatch and (2) the non-linear transconductance characteristics of the current accumulation bit line load cause current accumulation non-linearity, leading to the non-linear growth of the current accumulation line voltage. The current mismatch caused by these two non-linear problems leads to a large deviation in the value of the analog current/voltage and the ideal accumulated current/voltage transmitted to the lower ADC circuit, which eventually leads to the decrease in calculation digits and accuracy.

The HFIV circuit proposed in this work can solve the problem that analog non-linearity and limited dynamic range lowers the computational accuracy of analog CIM-SRAM. The HFIV circuit consists of a mismatch-free current mirror (MFCM) and a dynamic voltage-stabilized quantization (DVQ) circuit. For the MFCM circuit, as shown in Figure 9b, an operational amplifier (OA) is employed into a current mirror. The OA circuit can dynamically stabilize the drain voltage of the two transistors at the required voltage value, while ensuring the accurate replication of the current mirror to the accumulated current. It also can avoid the replication current mismatch caused by the voltage VRBL on the current accumulation bit line and the drain voltage mismatch of the current replication transistor. For the DVQ circuit, as shown in Figure 9c. The key of DVQ is to dynamically stabilize the voltage *V_X_*, based on the other OA, thereby reducing the system’s non-linearity greatly.

### 3.2. Linearity Analysis

#### 3.2.1. Static Operating Point Analysis of HFIV Circuit

Figure 10 shows the complete HFIV circuit. In the complete HFIV circuit, there are two voltage-stabilizing OAs. Through these two voltage-stabilizing OAs, *V_X_* and VBL/VBLB are stabilized at the reference voltage *V_REF_* of the OA. For MFCM, the drain voltage of the NMOS on both sides is fixed, so the accumulated current of the BL/BLB simulation calculation can be ideally replicated and amplified by the current mirror. Because of the fixed drain voltage, the non-linear drift of the replication current will not occur due to the increase in the number of calculation units. In the current replication and amplification stage, the MFCM circuit solves the non-linear accumulation problem of the calculation current. For the post-stage current–voltage conversion circuit DVQ, to obtain a stable *V_X_*, the circuit forms a negative feedback loop through a low threshold voltage LVTFET and OA2 to achieve the dynamic stability of the input voltage of the post-stage circuit. In the static state, due to the fixed voltage of the three nodes, the whole circuit can achieve linear accumulation of current and linear step output of voltage.

#### 3.2.2. Dynamic Voltage Regulation Analysis of MFCM

The post-stage current-voltage conversion circuit DVQ completes the dynamic voltage regulation of *V_X_* through LVTFET and OA2, as shown in Figure 11. In the small signal analysis, the *V_X_* value is not the complete level of static voltage, as shown in the figure. At point A in the diagram, when *V_X_* > *V_REF_*, the output voltage of OA2 will increase; that is, the gate voltage of LVTFET will increase. Because of OA2, the increase in gate voltage will be several times that of *V_X_*, so the current ID of LVTFET will increase accordingly, thus inhibiting the increase of *V_X_* and returning to *V_REF_* value (point B). After point B, when *V_X_* < *V_REF_*, the output voltage of OA2 will decrease, and *V_X_* will be dynamically stabilized on *V_REF_* by the same principle.

### 3.3. Behavioral Simulation

In the whole simulation of the CIM-SRAM, the HFIV circuit shows a linear voltage output, which solves the non-linear distortion problem of SRAM signals based on current-mode memory computing, as shown in Figure 12. In this simulation, the current accumulation of 0~64 units was calculated. In the first experiment, 32 SRAM units were simultaneously turned on for current accumulation. It could be seen from the results that the output voltage increased by 0.045 V (R = 5k) for each unit of calculated current reduction and obtained a linearity result of 99.92% with 0~32 SRAM bit-cells.

In the second experiment, 64 SRAM units were simultaneously turned on for current accumulation, as shown in Figure 13. It could be seen from the results that the output voltage increased by 0.015~0.018 V (R = 2k) for each unit of calculated current reduction, and a linearity result of 99.8% with 56~64 SRAM bit-cells was obtained. The number of cells, and the corresponding accumulated current and output voltage are listed in Table 1.

Through the global simulation of the MAC operation of the HFIV-SRAM array, the calculated current results and voltage conversion results of the HFIV-SRAM array in different modes of different rows were obtained, as shown in Figure 14. When the input data were 2 bytes, the calculation units were opened at the same time. The voltage conversion resistance set in this mode was 5 KΩ, and the current amplification ratio was 1:9. The simulation results show the linearity of the array simulation in this mode reached 99.973%. When the input data were 4 or 8 bytes, the voltage conversion resistance switched to 2 KΩ, and the current amplification ratios were 1:6 and 1:1, respectively. In these two modes, the excessive number of units opened at the same time led to a decrease in the linearity of the simulation calculation to 99.85% and 99.8%. 

Figure 15 shows a comparison between the HFIV-SRAM with some existing works. A T8T SRAM-CIM cell macro was proposed in [6] to support the multi-bit MAC operation of cnn, completing 1, 2, 4b inputs, 1, 2, 5b weights, and output of 7 bmax values by configurable reference voltages of different sizes. It reached 97.5% linearity in the simulated calculation. The CIM-SRAM macro with a linear compensation unit proposed in [8] showed good linearity in the calculation mode of a 64-bit input, up to 96.2%. The CIM-SRAM macro has a 64× binary (0 or 1) input, 64 × 128 binary (−1 or +1) weight, and 128 × 1-5-bit output. The column-level ADC converts the simulated point product result into an N-bit output code (N = 1 to 5), but 32 row units are used in the array for linear compensation, wasting a large calculated array area. 

## 4. Conclusions

After summarizing several calculation modes of SRAM and their respective advantages and disadvantages, this paper proposed a solution to the non-linearity of current-mode CIM-SRAM. The linearity of current-mode CIM-SRAM signals is not distorted by an HFIV circuit, and the working principle of the HFIV circuit is emphatically analyzed and expounded. In the simulation process, the CIM-SRAM with the HFIV circuit was used to analyze the linearity of the signal. The results show 99.92% (8~32 SRAM bit-cells) and 99.8% (57~64 SRAM bit-cells) of the linear output can be obtained after using the HFIV circuit to process the calculated signal. The process used was SMIC 55 nm. The simulation results showed the output accuracy of the voltage gradient quantization SRAM macro was 5 bit and the operating frequency was 200 MHz. Using the proposed circuit, the non-linear problem in multi-line SRAM units was solved. The HFIV circuit brings a very high linearity, but also produces a large circuit area and power consumption overhead. In future work, the HFIV circuit will be optimized to achieve an efficient compromise of the area, power consumption, and linearity of the calculation results.

## Figures and Tables

**Figure 1 micromachines-14-02180-f001:**
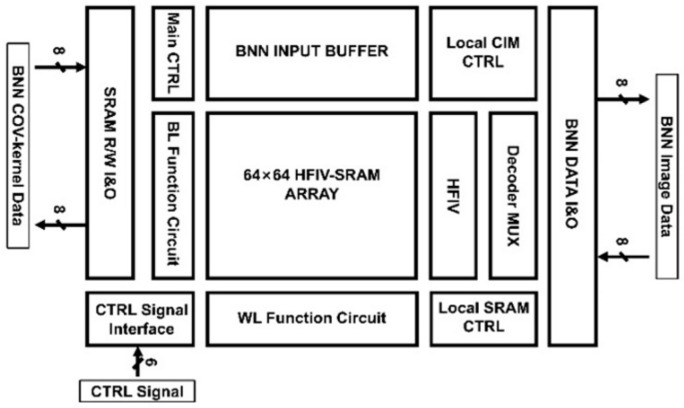
The overall architecture of HFIV-SRAM. The entire chip includes a 64 × 64 HFIV-SRAM array for the high-precision linearized HFIV circuit module, the data interface IO circuit and the corresponding control circuit for SRAM mode and CIM mode.

**Figure 2 micromachines-14-02180-f002:**
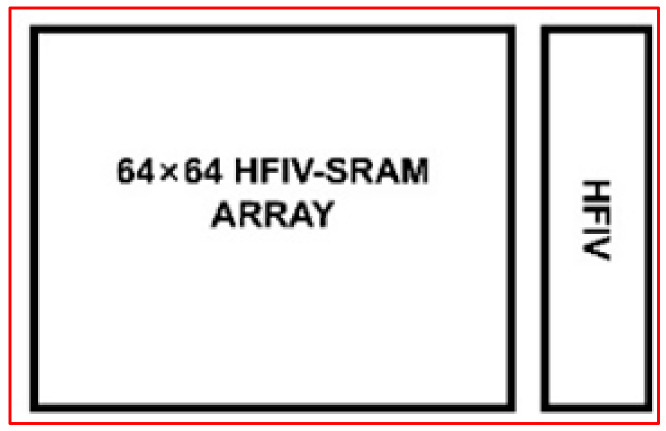
The HFIV-SRAM array and HFIV circuit block. After completing the CIM operation of the array, the simulated and calculated current is obtained, and the calculated current is processed by the HFIV circuit to obtain high-precision and high linearity results.

**Figure 3 micromachines-14-02180-f003:**
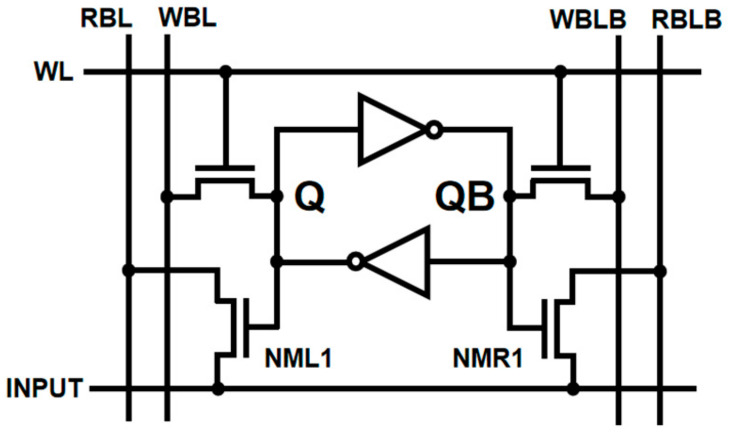
Typical 8T CIMSRAM cell. The 8T CIMSRAM unit consists of a conventional 6 TSRAM unit and two computing units, and the calculated current results are transmitted by RBL and RBLB to a digital-to-analog conversion circuit outside the array.

**Figure 4 micromachines-14-02180-f004:**
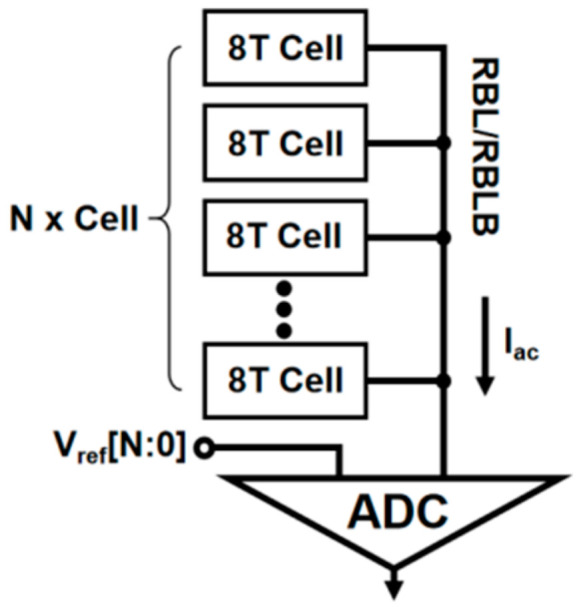
RBL’s current accumulate with ADC. Due to the non-linear spanning nature of the ADC load, the current IT accumulation on the RBL causes the serious non-linear problem of the calculation results when the number of cells n is large.

**Figure 5 micromachines-14-02180-f005:**
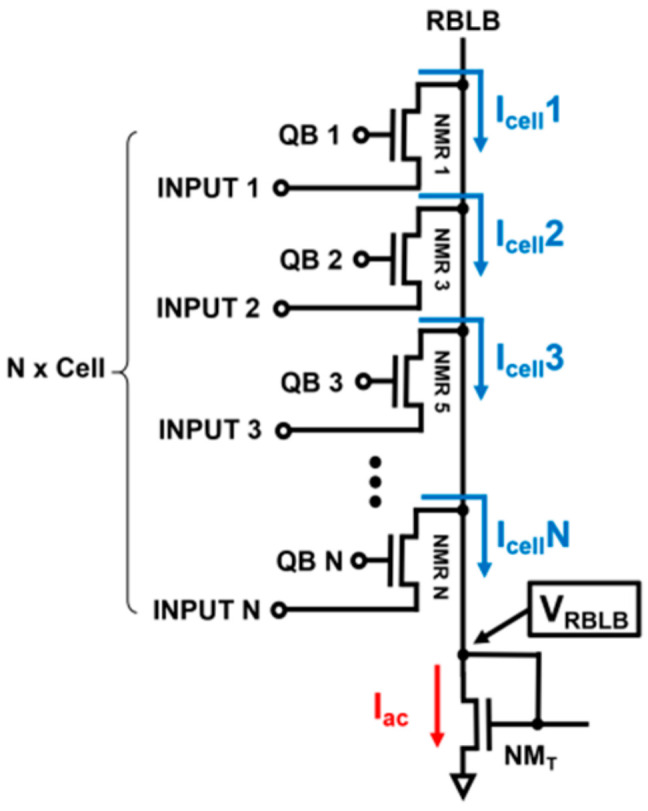
RBL with the diode load. When the number of units involved in the calculation increases, the cumulative current on the RBLB increases accordingly, causing the voltage of the node RBLB to rise, causing fluctuations in the unit current of a single calculation MOSFET NMR, which ultimately leads to non-linearity of the calculation results.

**Figure 6 micromachines-14-02180-f006:**
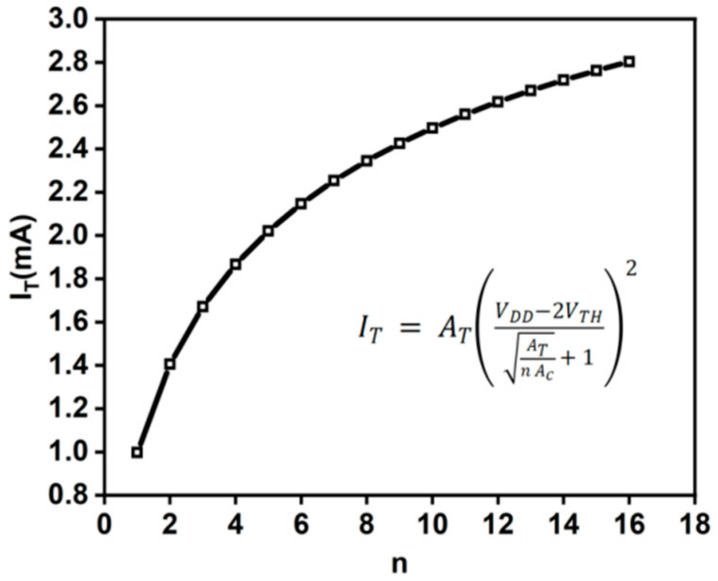
The non-linearity of *I_T_* on RBL. *I_T_* is the current result obtained by the CIM-SRAM array calculation. In the body silicon process of SMIC 55 nm, *V_DD_* is set to 1.2 V and *V_TH_* is 0.34 V. *A_T_* and *A_C_* are the process parameters.

**Figure 7 micromachines-14-02180-f007:**
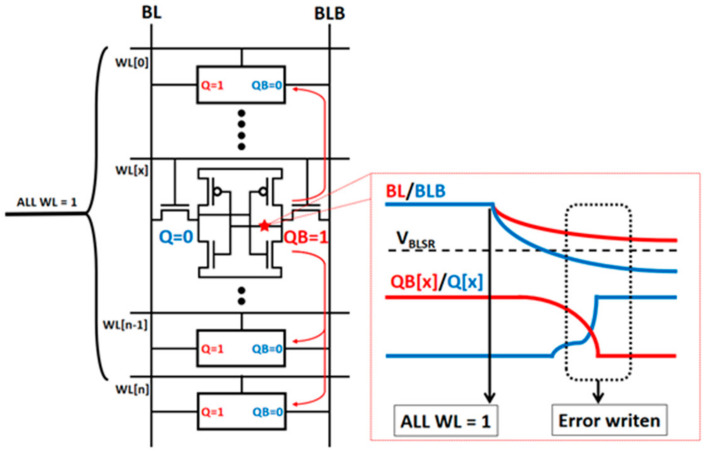
The phenomenon of read interference in a typical 6T CIM-SRAM. When the number of activated WLs causes the BL voltage to swing beyond the flip threshold voltage of the unit, read interference and data miswriting can occur.

**Figure 8 micromachines-14-02180-f008:**
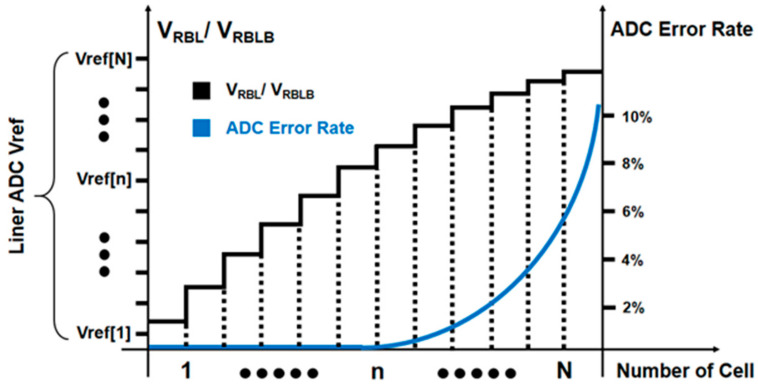
The non-linearity of *V_RBL/RBLB_* and the increase of ADC error rate. When too many word lines are activated simultaneously, the resulting RBL voltage drops, VBBL/VRBLB will be below the minimum voltage gradient that the ADC can accurately convert. At this point, the error rate of the analog conversion increases with the number of activation lines.

**Figure 9 micromachines-14-02180-f009:**
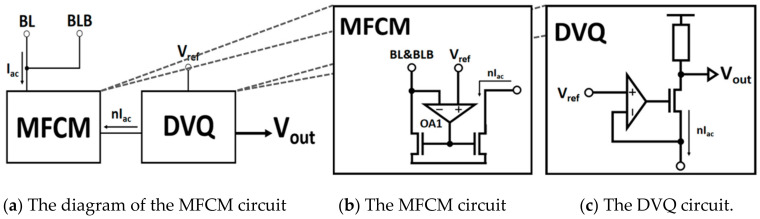
The high-precision fully dynamic range IV (HFIV) conversion circuit. The HFIV circuit consists of a mismatch current mirror (MFCM) and a dynamic stability quantification (DVQ) circuit. The OA circuit ensures the accurate reproduction of the cumulative current, while avoiding the replication current mismatch, thus greatly reducing the non-linearity of the system.

**Figure 10 micromachines-14-02180-f010:**
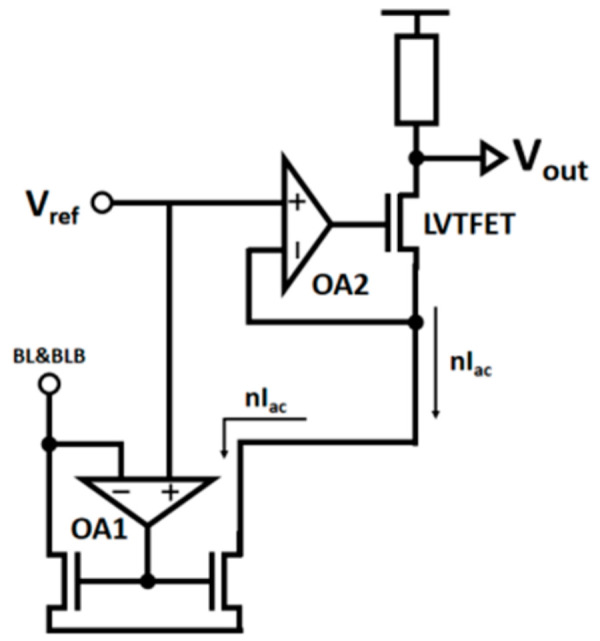
The HFIV circuit. In the HFIV circuit, two voltage stabilization OAs, a voltage stabilization current mirror, and a voltage stabilization current voltage conversion circuit are included.

**Figure 11 micromachines-14-02180-f011:**
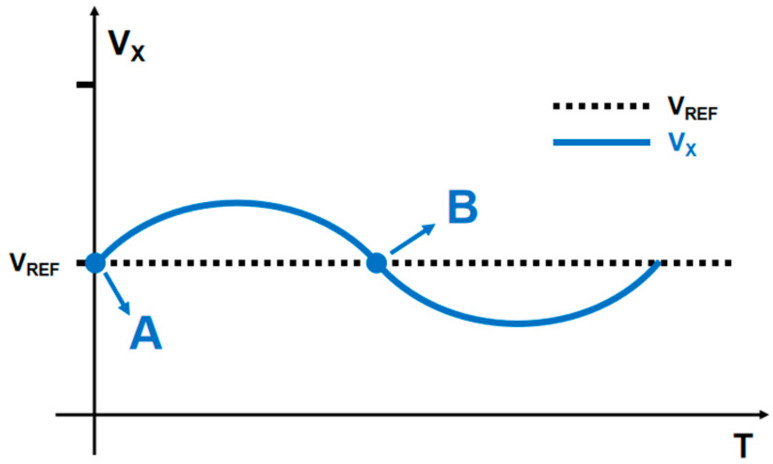
Dynamic voltage regulation of *V_X_*. Point A is that the output voltage fluctuation will be greater than the given calibration voltage. Point B is that the output voltage fluctuation will be less than the given calibration voltage.

**Figure 12 micromachines-14-02180-f012:**
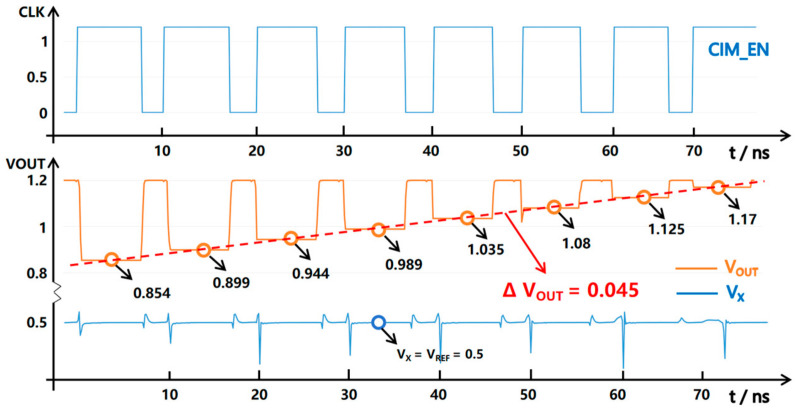
The behavioral simulation of the HFIV circuit with 8 SRAM units turned on for current accumulation. The figure shows when the number of SRAM units involved in the calculation decreases from 8 at-time clock cycles to 1, the voltage converted by the HFIV circuit shows extremely high linearity, and the voltage gradient stabilizes at 0.045 V.

**Figure 13 micromachines-14-02180-f013:**
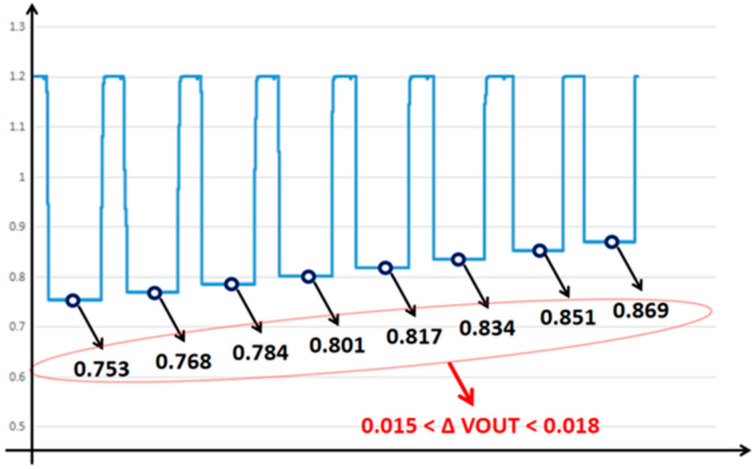
The behavioral simulation of the HFIV circuit with 56~64 SRAM units turned on for current accumulation. The figure shows when the number of SRAM units involved in the calculation decreases from 64 at-time clock cycles to 56, the voltage converted by the HFIV circuit shows extremely high linearity, and the voltage gradient stabilizes between 0.015 V and 0.018 V.

**Figure 14 micromachines-14-02180-f014:**
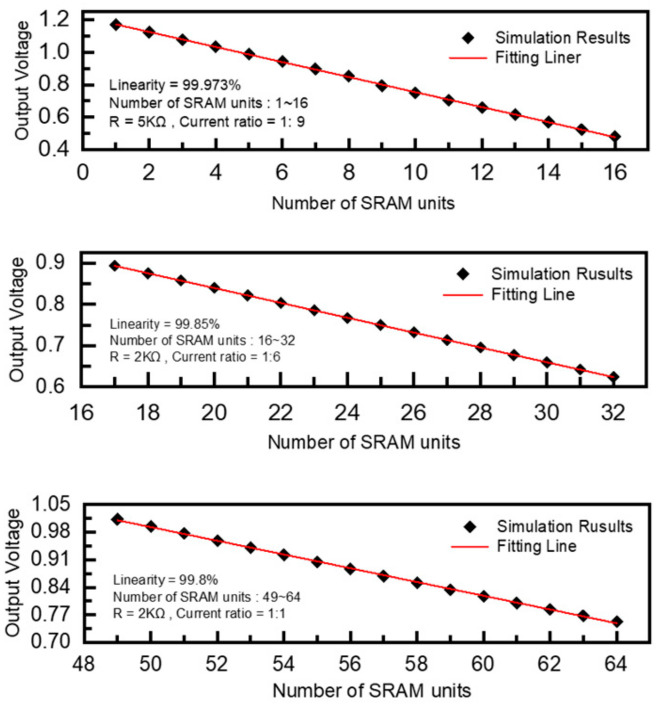
In the three modes with input data of 16, 32, and 64 bits, the calculated linearity of the HFIV-SRAM array was 99.973%, 99.85%, and 99.8%, respectively.

**Figure 15 micromachines-14-02180-f015:**
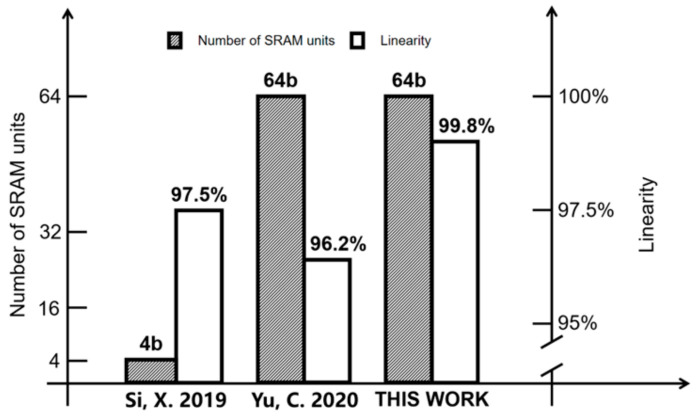
Comparing HFIV-SRAM against other existing works [6,8]. The comparison results show HFIV-SRAM not only calculates a high number of input digits, but also has extremely high computational linearity.

**Table 1 micromachines-14-02180-t001:** Number of SRAM bit-cells, accumulated current and output voltage for the second experiment.

Number of SRAM Bit-Cells	*I_ac_*/uA	*V_OUT_*/V
57	165.6	0.8688
58	174.3	0.8514
59	182.9	0.8342
60	191.4	0.8172
61	199.73	0.80054
62	207.92	0.78416
63	215.96	0.76808
64	223.58	0.75284

## Data Availability

Data are contained within the article.

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
