# Peer review of "A High-Precision Voltage-Quantization-Based Current-Mode Computing-in-Memory SRAM"

_micromachines, 2023, doi:10.3390/mi14122180_

Round 1

Reviewer 1 Report

Comments and Suggestions for Authors

The author proposes a solution to the nonlinearity of current mode SRAM for CIM applications. A HFIV circuit is proposed and investigated based on 55nm SMIC process, which achieves a calculation linearity to 99.92% and 99.8% at 1.2V supply. This paper presents sufficient results and discussion, which could be accepted after considering the following points.

 1. The English should be polished and avoid long sentences, which makes the Reader confused. The abbreviation like MAC should be provided a full spell at their first appearance, which should avoid reuse. Avoid “multiplication and addition MAC” in Page 2 Line 83.

Some unknown symbols appear at the beginning of the formula;

 2. Add some detail information to the caption of Fig. 1 to Fig. 10 and Figs. 12&13 are not clear enough, which are hard to follow for readers. 

 3. Do the results all come from simulation based on 55nm SMIC process? Is there any experiment data to support the idea and claim?

Comments on the Quality of English Language

The English must be improved.

Author Response

Dear reviewer, 
In response to your suggestions, I have made the following modifications to the paper:

1. The full text has been polished and grammatically corrected by the editorial department. Fixed incorrect symbols in formulas.
2. Specific descriptive information has been added to the titles of Figures 1 to 10, and more detailed data descriptions have been added to Figures 12 and 13.
3. The results in the full text are based on SMIC’s 55nm process. Due to laboratory funding issues, the flow chart has not yet been delivered for testing and measurement data will be collected in the future.

Reviewer 2 Report

Comments and Suggestions for Authors

The authors have proposed a  Voltage-Quantization-Based current-mode Computing-In-Memory SRAM that provides more linearity and consistency. The work clearly gives details on the issues and improvements over the present techniques. However, I recommend the following changes in the text:

1.    The highlight of the paper is linearity using voltage quantization. I would recommend having a graph showing the linearity and including more data with respect to the number of SRAM unit cells.

2.    While the authors have mentioned theoretically the problems with previous works, a comparative analysis with the proposed work with performance metrics like linearity, power, and area would give a complete understanding of the progress and improvements required in the proposed design.

3.    In the manuscript, the background is more as compared to the original work. I would recommend authors emphasize more on the proposed work than background. Please include more discussion and figures of your work to show the significance.

4.    The grammar in the paper needs improvement.

5.    The graphs need to be clearer.

6.    Please give the details of the simulation. If the work is simulated, the use of measured is misleading, please change.

Comments on the Quality of English Language

There are so many grammatical errors. Please correct them.

Author Response

Dear reviewer,
In response to your suggestions, I have made the following modifications to the paper:
1. In the chapter of Design and Analysis, a chart is added to show linearity and a comparative analysis with other existing work.
2. Additional discussions and descriptions of the proposed work were added.
3. The full text has been polished by English editors, and the icon content has been modified and more detailed description has been added.
4. The results of the full text are all based on SMIC's 55nm process. Due to laboratory funding problems, the flow sheet has not been delivered for testing, and the measured data will be collected in the future.

Reviewer 3 Report

Comments and Suggestions for Authors

This paper proposes a solution to the nonlinearity of current-mode CIMSRAM. The main question addressed by the research is the enhancement of memory performance by reducing the undesirable effect of non linearity thanks a specific arrangement. The proposed idea seems original and relevant according to the simulations. The way to deal with non linearity is specific. Simulations are probably not sufficient; a real test should be performed. 

It can be published after minor revisions.

Line 16, can increased > can be increased.

Line 49, is need > is needed.

Lines 112, 115, 117 and 120, the equations should have a number.

Line 120, the factor ( 1 + lambda VDS) is missing in equation 4.

Figure 6, numerical values of VDD, VTH, AT, AC are missing.

The numbers for equations are missing, the parameters in figure caption are not indicated.

Finally some typos should be removed in the text and in equation 4.

Author Response

Dear reviewer, 
In response to your suggestions, I have made the following modifications to the paper:
1. The full text has been polished by the editorial department and the grammar has been corrected. Fixed incorrect symbols in the formula.
2. Numerical labels are added to the equation.
3. For the factor (1 + lambda VDS) in formula 4, it can be eliminated in the calculation of formula (1), (2) and (3), so there is no silver in formula (4).
4. The results of the full text are all based on SMIC's 55nm process. Due to laboratory funding problems, the flow sheet has not been delivered for testing, and the measured data will be collected in the future.

Round 2

Reviewer 1 Report

Comments and Suggestions for Authors

The authors have revised the manuscript aftering considering most of my  questions.

Comments on the Quality of English Language

Can be improved.